# Understanding the Impact of COVID-19 on Chronic Lymphocytic Leukemia (CLL) Caregiving and Related Resource Needs

**DOI:** 10.3390/jcm12041648

**Published:** 2023-02-18

**Authors:** Diliara Bagautdinova, Kelsey C. Bacharz, Carma L. Bylund, Maria Sae-Hau, Elisa S. Weiss, Michelle Rajotte, Greg Lincoln, Taylor S. Vasquez, Naomi D. Parker, Kevin B. Wright, Carla L. Fisher

**Affiliations:** 1Department of Advertising, College of Journalism and Communications, University of Florida, Gainesville, FL 32611, USA; 2Department of Clinical & Health Psychology, College of Public Health & Health Professions, University of Florida, Gainesville, FL 32610, USA; 3Department of Health Outcomes and Biomedical Informatics, College of Medicine, University of Florida, Gainesville, FL 32610, USA; 4The Leukemia & Lymphoma Society, Rye Brook, NY 10573, USA; 5P.K. Younge Developmental Research School, University of Florida, Gainesville, FL 32601, USA; 6Department of Communication, College of Humanities and Social Sciences, George Mason University, Fairfax, VA 22030, USA

**Keywords:** COVID-19, caregiving, blood cancer, chronic lymphocytic leukemia, CLL, psychosocial impact, resource needs, spouse

## Abstract

Chronic lymphocytic leukemia (CLL) caregivers play a central role in disease management—a role that has been heightened during the COVID-19 pandemic given the healthcare system’s reliance on frontline family caregivers and CLL patients’ increased risk of infection and mortality. Using a mixed-method design, we investigated the impact of the pandemic on CLL caregivers (Aim 1) and their perceived resource needs (Aim 2): 575 CLL caregivers responded to an online survey; 12 spousal CLL caregivers were interviewed. Two open-ended survey items were thematically analyzed and compared with interview findings. Aim 1 results showed that two years into the pandemic, CLL caregivers continue to struggle with *coping with distress, living in isolation, and losing in-person care opportunities*. Caregivers described experiencing *increasing caregiving burden, realizing the vaccine may not work or didn’t work for their loved one with CLL, feeling cautiously hopeful about EVUSHELD*, and *dealing with unsupportive/skeptical individuals*. Aim 2 results indicate that CLL caregivers needed *reliable, ongoing information about COVID-19 risk, information about and access to vaccination, safety/precautionary measures,* and *monoclonal infusions*. Findings illustrate ongoing challenges facing CLL caregivers and provide an agenda to better support the caregivers of this vulnerable population during the COVID-19 pandemic.

## 1. Introduction

Chronic lymphocytic leukemia (CLL) is a blood cancer typically diagnosed in older adulthood [1] that involves accumulations of abnormal lymphocytes in the blood, bone marrow, and often the spleen and lymph nodes [1]. Some patients with CLL have a faster-growing form of the disease, and for others CLL can be slow-growing and treatment may not be needed for many years or at all. The chronic nature of CLL contributes to long-term biopsychosocial distress for both the diagnosed adult and their caregiver [2,3]. Managing CLL can involve lengthy and expensive treatments, severe treatment side effects, hospitalizations, relapse, and prolonged uncertainty [4,5,6]. CLL caregivers engage in a wide range of care tasks that can be physically, emotionally, and financially demanding [7]. A 2021 scoping review (which identified only one study [8] to date addressing the physical impact of CLL caregiving) showed that CLL caregivers reported poorer health-related quality of life compared to the general population [3]. In addition to disease-related physical effects and other co-morbidities common in older adults [9], patients often experience anxiety and/or depression [10].

While the COVID-19 pandemic has been challenging for cancer patients, patients with CLL and their caregivers have been particularly impacted. The nature of CLL prevents the immune system from functioning properly, increasing the risk of infections; in turn, CLL patients are at greater risk of getting very sick or dying from COVID-19 [11,12,13,14]. The COVID-19 pandemic further exacerbated caregiving burden as the healthcare system quickly became more reliant on “frontline family caregivers”; non-emergent medical visits were greatly reduced, and clinical protocols restricted visitors while the need for social isolation intensified to reduce infection risk [15].

Patients with CLL are also less likely to develop protective antibodies from COVID-19 vaccines [16,17,18]. Consequently, these individuals remain one of the most vulnerable populations impacted by COVID-19. Adults living with CLL and their caregivers have had to maintain social distancing precautions for longer than the general population [19,20]. 

Palliative, oncology, and chronic care experts have prioritized research aimed at better understanding the impact of COVID-19 on patients and their caregivers to develop resources and promote policy changes to better support them and minimize suffering [21]. In response to this charge, we previously published a study demonstrating the impact of COVID-19 on blood cancer caregivers of diagnosed aging parents at the start of the pandemic and global quarantine orders during the early phase (or wave one) and pre-vaccine period of the pandemic (i.e., conducted March–June 2020) [22]. We demonstrated that caregivers were grappling with several challenges, including increased isolation from loved ones (even at times from their diagnosed parent) and disruption in care due to reduced access to in-person care opportunities. Blood cancer caregivers described this period of quarantine as highly distressing, characterized by fear as well as uncertainty about COVID-19 risk.

Little has since been documented about the ongoing impact of COVID-19 on blood cancer caregiving as the pandemic continues into its third year. The ongoing impact on CLL caregivers continues to be of concern for patients as COVID-19 variants emerge. To help inform the development of needed services and resources, the present study aimed to (1) identify how the COVID-19 pandemic continues to affect CLL caregivers since the vaccine was deployed or the post-vaccine phase [23] and (2) identify COVID-19 related resources they needed during the pandemic.

## 2. Materials and Methods

This study is drawn from a larger mixed-methods study that included two phases: (1) an online survey to capture CLL caregivers’ experiences during the post-vaccine phase of the pandemic (disseminated February–March 2022), followed by (2) an in-depth, semi-structured interview with a subset of CLL spousal caregivers who responded to the survey (conducted April–May 2022). Inclusion criteria for the survey were (1) caregiver of a loved one living with CLL, (2) 18 years or older, and (3) currently residing in the United States/U.S. territories. Inclusion criteria for the interview were the same, with the exception that they be a spousal caregiver. Spousal caregivers were targeted for interviews given that most survey respondents were spouses. 

### 2.1. Recruitment

CLL caregivers were recruited in partnership with The Leukemia & Lymphoma Society (LLS) and CLL Society. Invitation emails for the online survey were sent to caregivers of a patient with CLL using existing LLS and CLL Society databases, reaching just over 15,000 caregivers. Survey participants were compensated with a $50 gift card and indicated their willingness to participate in an interview at the end of the survey when entering information to receive their compensation. Caregivers willing to participate in an interview were contacted via email after the survey recruitment ended. Interview participants received $50 for completing the interview. 

### 2.2. Measures

The online survey included demographic items, validated scales to capture caregivers’ psychosocial experiences and needs, as well as open-ended items. The present study analyzed two open-ended items that addressed experiences specific to COVID-19: (1) “How has the COVID-19 pandemic affected your experience as a caregiver?” (to address Aim 1); and (2) “what kinds of COVID-19 related resources and information would have been helpful to you?” (to address Aim 2). 

All interviews were audio recorded and conducted by phone by the senior author (CF) and lasted on average 76 min, ranging 56–90 min. Caregivers were asked about their psychosocial experiences and needs relating to CLL in general. The impact of COVID-19 came up organically during interviews as caregivers described their CLL caregiving experiences. To address Aim 1, additional probing questions were utilized during the interviews to further explore the impact of the pandemic on their lives. Interviews were professionally transcribed resulting in 240 pages of data.

### 2.3. Analysis

A deductive and inductive thematic analysis was conducted on both the survey open-ended responses and interview transcripts using a constant comparative method (CCM) approach [24,25]. As Aim 1 was answered with both survey data and interview data, analyses were kept separate to triangulate (i.e., compare) the findings from each dataset to enhance rigor in the results (i.e., interview findings validating survey findings) and to capture distinct findings by method. Deductive (i.e., closed) coding first informed the analysis, using our previous study findings as a priori themes that represent challenges experienced during quarantine (i.e., fears and uncertainty or distress; isolation; reduced in-person care) [22]. Inductive (i.e., open coding) was conducted in conjunction with deductive coding to capture new themes not identified in our previous study. 

Analysis for both Aims 1 and 2 contained the following four analytical steps from a CMM approach. Thematic analytical steps of CCM for data addressing both Aims 1 and 2 included (a) immersing oneself in the data by constantly reading and comparing participants’ responses; (b) identifying concepts and assigning codes (i.e., labels) to text in responses (i.e., open and closed coding); (c) grouping codes into categories that represent patterns in the data or emergent themes; and (d) conducting axial coding of each theme to identify characteristics (i.e., thematic properties) to better define each theme (see Figure 1). Frequencies were determined for survey responses given all respondents received the same questions [26]. A widely used set of criteria to determine thematic saturation was used (i.e., noting repetition, recurrence, and forcefulness in responses) [26]. In line with best practices for interpretive study designs (i.e., not all participants received the exact same questions) [26,27], frequencies are not reported with interview data and rather the extent of saturation is provided [28]. All themes reported were confirmed with at least a saturation of 42% of caregivers interviewed, with the majority of themes saturated by 58–75% of caregivers interviewed.

Multiple coders were used to ensure rigor in the analysis process [29]. One author (D.B.) led the analysis of the survey data for both aims and developed separate codebooks for each aim. A second author and qualitative expert (C.F.) reviewed all analyses individually to come to a consensus and further refine both typologies/codebooks. Another author (T.V.) used a closed coding approach to then validate (i.e., confirm) the final typologies of each codebook in a subsample of survey data for both aims. Analyses of the interview data were initially conducted by one author (C.F.) concurrently with data collection to develop a preliminary codebook to ensure rigor and thematic saturation. The preliminary typology was then validated and extended by the second author (K.B.) who also identified thematic properties with analysis, overseen by C.F., to concurrently update and refine the codebook. 

For Aim 1, the senior author compared the finalized codebook from the interview data with the survey findings codebook to identify similar and divergent findings for presentation. Table 1 was developed to present Aim 1 findings using the ecological sentence synthesis approach to ensure findings are easily translatable into resources [30]. Findings for Aim 2 are presented in Table 2 to illustrate resources caregivers reported needing during the pandemic with frequencies included to capture the most prevalently reported resources.

## 3. Results

### 3.1. Survey Participants

A total of 575 caregivers responded to the survey. Of the respondents, 377 were women (65%) and 195 were men (34%), with 88% of those in a married/domestic partnership. Caregivers ranged in age from 18 to 63 (*M* = 47.66; *SD* = 10.32). Most caregivers identified as white race (91%) and having at least one child (79%). Additionally, 41% had less than a four-year degree. Half of the participants stated they were retired, and 23% were employed full time. Caregivers reported residences within 45 different U.S. states. 

Most caregivers reported caring for a spouse (82%). The majority of caregivers’ diagnosed loved ones had current or previous active treatment experiences: 50% were currently in treatment, 20% had completed treatment and were in remission, 5% were about to start treatment, and 1% had elected to stop treatment but were not in remission. An additional 24% of caregivers’ diagnosed loved ones had not yet had treatment and were being monitored (i.e., in watch and wait status). Caregivers’ diagnosed loved ones were diagnosed with CLL between 1 and 22 years prior and, on average, 59 years old when diagnosed (range = 42–86 years old). A total of 550 caregiver respondents answered the open-ended item about the impact of the pandemic (Aim 1). Of these, 128 (23%) were in watch and wait (95% of caregivers of loved ones in watch and wait responded to the open-ended item). A total of 473 survey respondents answered the item on desired resources (Aim 2). 

### 3.2. Interview Participants

Twelve spousal caregivers (10 wives and 2 husbands) were interviewed. Caregivers (M_age_ = 63, range = 50–72 years) and diagnosed spouses (M_age_ = 67, range = 49–79 years) were in midlife to later adulthood and primarily white race, with one caregiver identifying as Hispanic. Caregivers’ spouses were diagnosed between 1 and 22 years prior and received care in 11 different states in addition to Washington, DC. Five caregivers and one diagnosed spouse were employed, whereas the majority were either retired or not working. Caregivers predominantly described themselves as financially secure, with four caregivers describing notable financial struggles. Most reported not having any children (*n* = 7). All of the caregivers’ spouses were either currently in treatment or had been in treatment.

### 3.3. Aim 1: Challenges CLL Caregivers Encountered during the Pandemic

Both survey and interview responses demonstrated that regardless of treatment status, caregivers of patients with CLL report the same three challenges encountered during quarantine [22] persisting more than two years since the pandemic began: coping with distress (i.e., fear and uncertainty), living in isolation, and losing in-person care opportunities. Survey and interview findings also captured additional struggles of increasing caregiving burden, regardless of treatment status; however, this challenge was more prevalent with caregivers of diagnosed loved ones with current and/or previous treatment experience. Interview findings identified three unique challenges related to living with CLL during the post-vaccine deployment phase of the pandemic: realizing the vaccine may not work or did not work for their loved one with CLL, feeling cautiously hopeful about EVUSHELD (monoclonal antibody infusions), and dealing with unsupportive or skeptical individuals (see Table 1).

Interview findings both validated and extended survey results. As such, survey and interview results for Aim 1 are reported collectively. Themes are illustrated using caregivers’ written or spoken words. Properties of themes (in italics below) are provided to further characterize caregivers’ experiences.

#### 3.3.1. Coping with Distress 

Caregivers responding to the survey (*n* = 223) as well as caregivers who were interviewed collectively described grappling with chronic distress. This included survey respondents caring for loved ones in watch and wait (*n* = 60). Caregivers wrote about living with “great fear” and “anxiety” with responses illustrating distress, focused either on a state of heightened caution to reduce risk, increased negative feelings, or COVID-19 uncertainty-related distress. Caregivers wrote about being “paranoid of bringing infection home” (Survey-91), which warranted their vigilance in maintaining heightened caution to reduce their (and their loved one’s) risk of COVID-19 infection (e.g., shopping at certain times, double masking, and wearing gloves). They wrote that “the pressure was real” and used language that captured their increased negative feelings, for example, “feeling paranoid”, “crazy”, and “nervous and anxious”. They also described distress related to uncertainty about COVID-19 risk: “More stress due to uncertainty about my husband’s safety”.(Survey-11)

Caregivers’ reports in the interview study validated this finding, illustrating the negative effect they endured: “Our lives have been sheer hell during COVID, because he’s immunocompromised” (Interview-7). Multiple caregivers expressed that coping with the pandemic (i.e., maintaining caution) was more strenuous than coping with CLL: “It’s scary. I was very, very protective about COVID because I’m thinking, ‘I don’t need you to die of COVID. We’re getting through this cancer!’” (Interview-10). They also linked their fears and heightened caution with inhibiting their mental well-being: “Quite a bit of depression on my side. I worry about her quite a bit. … I didn’t want to bring something home, and so I ended up with quite a bit of anxiety over these things and her”. (Interview-12)

Caregivers who were interviewed also provided more characterization of uncertainty distress about COVID-19 risk by highlighting two issues. First, COVID-19 uncertainty inhibited their ability to make future plans for things, such as family/social events, which, in turn, impacted their long-term vision of the future:

How are we going to live in this world or how do we want to live in this world? How many risks? Do we want to get on a plane? … Are we going to be able to go take our trips like we would like or go to someone’s wedding like we won’t do at this point? That kind of overlays because of the fact that he’s such an at-risk individual for COVID hospitalization and death. It’s sort of like, “What do we want to do here?” The CLL seems—not to be trite—like small potatoes right now. He’s on this new drug [which] makes his white count go up for a while and then down. … It’s more like “Can you stay well through COVID? Can we still have a good life?”.(Interview-8)

Caregivers also shared how their uncertainty about risk did not always mirror their diagnosed spouse’s perception and, as such, they managed different risk perceptions and uncertainty. While risk perceptions evolved across the pandemic, caregivers tended to prioritize their diagnosed spouse’s perceptions: 

I treat myself the way she wants to treat herself. That’s part of what I can do to support her. I’ll wear maybe a surgical mask, which I don’t really know if you really have to if you’re outside. But if we’re walking and we come near somebody, even if we’re outside, she’s telling me, “Put your mask on!” So, doing things like that makes her feel more confident even though I’m not really sure you need to wear a mask outside.(Interview-9)

However, as COVID-19 risk and related restrictions decreased after the vaccine was deployed, caregivers’ COVID-19 related uncertainty shifted, though not always in line with their diagnosed spouse’s, which might pose a new challenge in the near future: 

Now that things are lifting, I’m definitely feeling much more frisky, I guess I would say, and [he’s] not at all. He really does enjoy the safety and the predictability of staying home. So how we navigate that, I think, is going to be the next issue.(Interview-2)

#### 3.3.2. Living in Isolation

Isolation was a persistent challenge as reported by both survey respondents (*n* = 200) and caregivers who were interviewed. This included survey respondents caring for loved ones in watch and wait (*n* = 33). Caregivers’ survey responses either addressed the pandemic impact generally in terms of “living in a bubble” or in reference to what they lost due to isolation: losing social time/connection with family/friends or losing social activities. Caregivers wrote about living in a “bubble” using language that linked their isolation with distress (e.g., “horrible”, “heart-breaking”, and “overbearing”). They described the social consequences of the pandemic both in terms of losing intimate connections (e.g., “The Covid pandemic has totally isolated my wife and I from friends, family and everything we enjoy in life”. Survey-323) and social activities: 

We see no one except outside, and since we live in the Midwest, the weather means that most of the year we have a social life only on Zoom. We don’t travel, go to concerts, see films in theaters, dine in restaurants, get our hair cut professionally. In fact, because the pool in our community is physically attached to a care center, we have not used the pool since the Delta surge began. (Survey-316)

These collective social losses negatively affected caregivers’ well-being: “It’s been isolating for me, because whomever I’m exposed to, she’s exposed to. I’m very social and it’s been difficult not to see my friends and family as frequently as I used to prior to the pandemic”.(Survey-411)

Caregivers’ reports in the interview study validated this survey finding, with caregivers noting the impact of losing social activities, such as travel, going to church, eating out at restaurants, and going to concerts or community events with friends or loved ones:

Because of COVID we can’t have social activities. Both she and I were in our church choir, and we can’t do that anymore. So, one of the biggest, I’d say, challenges for me is you know, a lack of social contact and we had always gone to church every Sunday and we can’t, you know. We can’t.(Interview-9)

Caregivers in the interview study also contributed new insights as they described employing strategies to cope with isolation, to both promote social connection and enhance their well-being. They used technology (e.g., Zoom) to interact more with loved ones or moved social activities outside when weather permitted. Caregivers in warmer climates expressed gratitude for having social opportunities given they could be outside, with at least one caregiver moving from a colder to a warmer climate after diagnosis: 

We would go out and spend the afternoon out by our pool, have snacks and stuff and go swimming, go to the beach, go fishing. There’s just so much available at least here to keep us from completely going out of your mind. (Interview-1)

Also, unlike early in the pandemic, these caregivers now had access to rapid COVID-19 tests, which some described using to feel comfortable visiting loved ones: “We take great faith in the rapid tests. We’ve just now started going to dinner at friends’ houses and if they take a rapid, we’re good”.(Interview-10)

#### 3.3.3. Losing In-Person Care Opportunities

Caregivers struggled with lost in-person care opportunities across the pandemic as reported in the survey study (*n* = 71) and interviews. While this did include some survey respondents caring for diagnosed loved ones in watch and wait, it was less prevalently reported as a challenge (*n* = 11). Survey respondents focused on two types of lost in-person care. They addressed both reduced in-person clinical care opportunities as well as lost in-home help. These losses were also associated with distressing language and isolation, with caregivers disclosing they felt “frustrated” and “cut off”. Caregivers were still at times unable to attend appointments either due to clinical protocols or COVID-19 exposures and infections, which could inhibit their ability to provide care: “[It’s been] extremely difficult to navigate … cannot attend appointments. [My husband] missed several biopsies etc. due to not having me attend and not understanding his treatment plans or what was expected and getting lost” (Survey-480). Caregivers also described losing in-home care help: “[COVID-19] made it difficult to get outside assistance”.(Survey-147)

Caregivers in the interview study validated this finding, describing the same two areas of impact. They also extended survey findings about reduced in-person clinical care opportunities by noting appointment challenges, either with getting appointments or their dislike of telehealth: “Because of COVID and if you do want to go see a doctor it’s like, ‘We don’t have an appointment for seven months,’ or, ‘We can do something on Teams or on Zoom.’ But it’s like me, No! I need to talk, and I want to see you” (Interview-3). Caregivers also described losing in-person support opportunities for themselves: 

As a newly diagnosed couple … it wasn’t all that easy to find resources. And it wasn’t all that easy to take it all in … It was harder to find because it all happened during COVID. And so when you look at a support group or something like that, it was all online or postponed. … You feel like you’re being barraged with all these really intense decisions, and you just don’t have the skills, the knowledge, or the ability to take it all in fast enough. It’s coming at you so quick. And the learning curve is so steep.(Interview-5)

#### 3.3.4. Increasing Caregiving Burden

In addition to the ongoing challenges caregivers faced since quarantine, two years into the pandemic, caregivers who responded to the survey (*n* = 42) and interview also reported significant caregiving burden, which included increased financial burden and instrumental caregiving burden. Although notably less prevalently reported, caregivers of loved ones in watch and wait did report the same types of burden (*n* = 7). For instance, caregivers wrote in their survey responses about the financial strain they endured because of COVID-19 given their spouse was “forced to retire early”, “not working due to her condition”, or “fired from his job because of COVID due to him being immunocompromised”. They described how their spouse’s loss of employment resulted in extreme financial strain including “los[ing] our main source of income”, “nearly cost us our little mom & pop shop”, and “had to file bankruptcy, loss of condo, loss of finances”. Caregivers also disclosed financial insecurity because they themselves had “reduced opportunities to earn more money, needing to avoid close contact with others”. They also encountered financial strain related to getting COVID-19:

My husband contracted COVID and almost died. He was hospitalized for a month and received home health care for another month. He’s developed other problems with his heart and breathing. He currently is on long term FMLA from his job and receiving unemployment. I have to pay his insurance premiums to maintain his insurance. I’ve tried submitting these claims for reimbursement but get denied. … It’s been a huge financial burden cause it’s quite expensive. Hard on finances.(Survey-329)

Caregivers also reported instrumental care burden, which was related to their need to isolate and not receive in-person or at-home care help to reduce risk: “[COVID-19] has led to me taking on more outside tasks to minimize his exposure” (Survey-502). They juggled most if not all of the household responsibilities (e.g., keeping the house clean, outside maintenance) as well as outside tasks (e.g., grocery shopping). They wrote about solely carrying this load: “I have carried on my shoulders, the duty of protection, supplier of needed items” (Survey-507). At times, the instrumental care burden intersected with financial burden: “The sole caretaker for the entire household, while working a full-time job”. (Survey-314)

Caregivers in the interview study validated this finding, describing both financial and instrumental caregiver burden. Although most caregivers who were interviewed acknowledged they were economically secure or already retired, they still needed and sought financial resources for medications or treatment from organizations, such as LLS and the CLL Society. A smaller subset of caregivers also disclosed notable financial strain:

The financial blow is the bigger one for me and for both of us. … Our issues are largely around how are we going to tread water. … Normally, he would be going back to [work] but the stress, the exposure, the difficulty on the body, we’re doing everything in our power not to make him go back. … So, we’ve sold our car. … I work a side gig. And my parents—my family—has come together in tremendous ways to support us financially.(Interview-5)

During interviews, caregivers also expressed how overwhelming and persistent their instrumental care duties were: “I don’t think a week goes by without something else, needing a doctor’s call or fatigue, bruising…” (Interview-2). Those providing care and working full time noted their extreme burden, which impacted their mental well-being: 

I was juggling so much. I mean, it was overwhelming. It was truly overwhelming. … I don’t really want to be going out in the public and bringing something home to my immunocompromised husband. … Then it’s like, what? Do you hire somebody to come in, and then how does that get paid for? So, yeah! That’s been a big part of it, at least the stress part of it, like trying to juggle a 40-plus-hour-a-week job and then working around the appointments and having a bad night but then having to get up and get on the phone in the morning and dealing with that. Oh my gosh. It was making me crazy. It really was. … I did start seeing a therapist on the side. I did that for myself once a week for I don’t know how many weeks I did that—for months, actually.(Interview-1)

#### 3.3.5. Realizing the Vaccine May Not Work or Didn’t Work for Their Loved One with CLL 

Although not reported in the survey, caregivers who were interviewed also described an experience after the vaccine was deployed during the pandemic that was specific to CLL patients—facing that the vaccine did not protect their spouses from COVID-19. They characterized this experience in two parts. They initially held an immense sense of hope and excitement about the vaccine: “We thought that was going to change everything” (Interview-2). Caregivers even described how they personally made strong commitments to the vaccine, which included traveling long distances to receive it and making lifestyle changes to enhance efficacy: “My husband lost 65 pounds on purpose the first year, because I said, ‘You have to get in shape for the vaccine. The chance of you responding to these vaccines is not high, but it’s really even lower if you’re overweight’” (Inteview-7). Once their spouse was vaccinated, caregivers shared their shock and devastation when then learned their spouse did not develop an immune response. They recalled feeling “a real shock”: “We were so excited to go and get the shot and then he was so crestfallen to learn that he had produced no antibodies”.(Interview-8)

#### 3.3.6. Feeling Cautiously Hopeful about EVUSHELD

Additionally, caregivers who were interviewed shared their cautious hope about the promise of monoclonal antibody infusions, specifically EVUSHELD, to protect their diagnosed spouse from COVID-19. They characterized this as jointly experiencing two distinct emotions that encapsulated a sense of cautious optimism. They described EVUSHELD renewing hope about spouse’s protection from COVID-19 but, at the same time, being cautiously hopeful and maintaining precautions. One caregiver described the juxtaposition as EVUSHELD was both “gold” and, simultaneously, “not a silver bullet”.

He got the monoclonal antibodies just a few weeks ago. That’s like gold. It’s hard to get it. You can’t find it anywhere and he had to be invited, but luckily, they did. I guess his hematologist put a message through and he was invited to get that. So, he just did now, but it’s not a silver bullet. It’s a little bit of a breather. So, we don’t have to maybe be quite as worried. Again, not a silver bullet that you just go out and go—okay! Now, we can go out and do all this stuff.(Interview-1)

Caregivers explained that they tempered their expectations given the devastating experience learning the vaccine did not work and the unpredictable nature of the pandemic: 

We’re a little gun-shy because we thought as you said once the vaccines came out we’d be good. We’d all be good to go and then Omicron came. And I mean Shanghai is closed down. It’s crazy the stuff that still goes on.(Interview-2)

#### 3.3.7. Dealing with Unsupportive or Skeptical Individuals

Caregivers who were interviewed also shared how as the pandemic progressed, they encountered distressing experiences, often with family and friends, who did not support their precautions to protect their spouse’s health. They shared that, at times, they had to confront divergent risk perceptions/behaviors, which contributed to personal distress, isolation, and relational tension. They disclosed both indirect and overt experiences with individuals whose perception of risk were not in accord with their own needs: 

She used to go walking with some friends of hers just as a little exercise walk, a little socialization. And they didn’t understand. … that she would have to wear a mask if she’s close to them or she’d want to walk six feet away from them. And they didn’t say anything negative but she just got the feeling that, “Oh, they really don’t want to walk with me anymore”.(Interview-9)

At times, caregivers encountered this divergence as well as skepticism outside their social network, which included at least one caregiver sharing an overtly hostile experience while shopping in a store:

We both still wear our masks when we go into a store, and sometimes people are quite rude about things. I had a man come up to us in this store, and he says “I don’t know why you both have a mask on, they’re not doing any good”. He was with his wife, and he was going around harassing people, and I said “Does your wife have cancer? Or hopefully she doesn’t. But my wife has cancer, and we just can’t take a chance on things”. It’s kind of disturbing when people don’t really understand things because they don’t have a problem, and, to me, that’s kind of rude of people. I mean, nobody understands things until it happens to them.(Interivew-12)

Conflicting risk perceptions also contributed to family members’ unwillingness to accommodate their need for precautions:

Among his family, there were a lot of anti-vaxxers, and even a couple of folks who lean a little bit too QAnon into the “plandemic”. And so that was a huge issue. It’s like, wait a minute, you expect us to come to all these family events. You’re going to hold them indoors. You’re going to shove 12 people in the tiniest room in the house, where you’re then going to serve birthday cake with candles on top, have the birthday person blow on the cake, and then serve it to my husband with the immunocompromised [system] … Check, please. I mean, I couldn’t get out of the house fast enough. I was aghast.(Interview-5)

This lack of support and understanding impacted their ability to socialize, and when it involved loved ones, unsupportive behavior or skepticism contributed to tension and pain: “They’re anti-vax and they’re anti-mask and you know, it’s kind of hard. They won’t do anything to be accommodating to my wife. And so that really hurts”. (Interview-9)

### 3.4. Aim 2: COVID-19 Resources Caregivers Needed or Wanted during the Pandemic

Table 2 presents results from the open-ended survey item addressing Aim 2, in which 473 caregivers were asked what “would have been helpful” to them during the pandemic. Of these, 48% (*n* = 227) reported having had all the resources they needed during the pandemic. The remaining 52% (*n* = 246) reported needing information and access to resources in four categories (COVID-19 risk, vaccination, safety/precautionary measures, and monoclonal infusions) and that the information be ongoing and reliable or from trusted sources.

## 4. Discussion

This study examined the self-reported experiences and resource needs of caregivers of patients with CLL two years into the COVID-19 pandemic. Findings demonstrate that CLL caregivers continue to endure strenuous challenges, including distress, isolation, and reduced in-person care. Our previous study showed these are three challenges caregivers began enduring at the very beginning of the pandemic (e.g., March–May 2020) [22]. Thus, these challenges have persisted for more than two years. Findings also reveal new stressors encountered after the vaccine was deployed or the post-vaccine phase, including coping with insufficient antibody responses to vaccination as well as unsupportive or skeptical individuals. Family caregivers play a critical role in the management of CLL and, collectively, these findings inform an agenda for resource development to better support CLL caregivers and their diagnosed loved ones.

### 4.1. Reducing Caregivers’ Distress/Burden by Attending to Their Risk Information Needs 

Findings illustrate that COVID-19 related distress continues two years into the pandemic as CLL patients and their caregivers continue to remain isolated and protective, while often being misunderstood and questioned by friends, loved ones, and even strangers. Frontline family blood cancer caregivers in both the survey and interview studies described a “living hell” in that they were overwhelmed with emotional distress characterized as living in chronic “fear”, “paranoia”, and “anxiety” that their loved one with CLL would contract COVID-19. Relatedly, caregivers disclosed fearing they may be the one to expose their diagnosed loved one to COVID-19 and, ultimately, be responsible for putting their life at risk. Their emotional distress was made worse by the need for precautions that require social isolation, as isolation is also widely known to be detrimental to psychological well-being [31]. 

In turn, socially isolating to reduce risk also means they cannot accept caregiving help (e.g., respite care and in-home help with household tasks), thereby furthering caregivers’ burden—another known factor that is linked to poorer psychological well-being [32]. As caregivers themselves pointed out, the pandemic is woefully unpredictable with new variants and new resistance consistently changing the landscape of risk. It is also noteworthy how the failure of the vaccine in adequately protecting the CLL community further contributed to caregivers’ distress as they accepted the reality that their heightened risk, need for caution, isolation, and distress would continue for the unforeseeable future.

Ongoing care burden and distress can inhibit caregivers’ sense of control and ability to cope or engage in caregiving [33] and is also associated with less openness between caregivers and their diagnosed loved one [34]—effects that are all detrimental to disease management. CLL caregivers in the survey study indicated they need ongoing, reliable information from experts on COVID-19 risk that is specifically targeted to CLL patients. Caregivers also reported wanting information on how to access equipment or resources critical to their ability to reduce COVID-19 risk (e.g., access to monoclonal antibodies or masks). Given their elevated uncertainty about COVID-19 risk and precautionary needs, this information may help reduce caregivers’ distress and burden, while enhancing their ability to make precautionary decisions. 

### 4.2. Addressing Divergent Risk Perceptions by Providing Interactive Information Tools

CLL caregivers who were interviewed also shared struggles navigating different risk perceptions, both between themselves and diagnosed spouse and between themselves (caregiver–spouse) and other individuals, typically loved ones. As the pandemic has evolved and COVID-19 risk has decreased after the vaccine rollout, caregivers acknowledged that their risk perceptions are sometimes not as heightened as their diagnosed loved ones’ perceptions. To cope, they have prioritized their loved one’s preferences. However, some caregivers acknowledged now struggling with maintaining the same isolation precautions—an issue CLL caregivers and diagnosed loved ones may need help discussing and negotiating in the near future. Additionally, caregivers described painful relational challenges sparked by divergent risk perceptions with loved ones, including a refusal to respect their need for precautionary measures. Caregivers described sometimes tumultuous situations with family who questioned the existence of COVID-19 or were described as “anti-vax” or believers of online conspiracies about the pandemic.

As they described their resource needs during COVID-19, caregivers expressed needing advocacy or resources to help explain risk to others (such as family) who “do not understand” their unique, life-threatening situation. Such resources may help caregivers navigate challenging conversations with skeptical individuals, particularly within their social network where they need ongoing support. The relational implications of COVID-19 on more vulnerable populations, including CLL patients and caregivers, have yet to be explored, with only a small number of studies illustrating the rise in family conflict and disruption related to pandemic skepticism [35,36,37]. As more at-risk populations continue to contend with life-threatening COVID-19 risk, more widespread education is needed to elevate awareness on a societal level about the realities of their vulnerability and cautionary needs. At the same time, interactive resources (i.e., conversation tools or communication skills training) from expert sources may help caregivers navigate challenging conversations with unsupportive loved ones that may be critical to maintaining family ties and obtaining social support. 

### 4.3. Limitations

This study was only conducted in the U.S. with the majority of participants identifying as white and non-Hispanic. The lack of racial and ethnic diversity reflects the fact that CLL is much more commonly diagnosed in men (i.e., 60% of those diagnosed are men) and in individuals who are white (90% of CLL patients are of white race). However, cultural diversity both within the U.S. and across the globe will impact caregivers’ experiences and needs. Additionally, the majority of individuals in the interview study did not have children, which may have impacted their experiences and perceptions. 

## Figures and Tables

**Figure 1 jcm-12-01648-f001:**
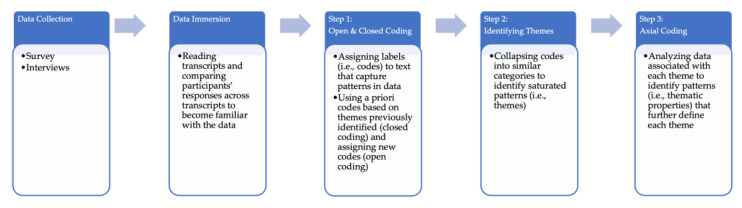
Thematic analytical steps of CMM.

**Table 1 jcm-12-01648-t001:** Challenges Encountered by CLL Caregivers During the COVID-19 Pandemic.

CLL Caregivers Struggled with These Challenges	Characterized by
[since the pandemic began]	
Coping with distress	A sense of heightened caution Increased negative feelings (e.g., fears) Covid-19 related uncertainty (which inhibited their ability to make plans and sometimes resulted in different risk perceptions/uncertainty than spouse)
Living in isolation	“Living in a bubble” Losing social time/connection with family/friends. Losing social activities Enacting coping strategies (e.g., rapid tests, outsideactivities)
Losing in-person care opportunities	Reduced in-person clinical care opportunities (and associated appointment challenges) Lost in-home help
Increasing caregiving burden	Increased financial burden Increased instrumental caregiving burden
[after the vaccine was deployed]	
Realizing the vaccine may not work or didn’t work for their loved one with CLL	Hope and excitement about the vaccine Shock and devastation when with no immune response
Feeling cautiously hopeful about EVUSHELD	Renewing hope about spouse’s protection Being cautiously hopeful and maintaining precautions
Dealing with unsupportive or Skeptical individuals	Divergent risk perceptions/behaviors Family’s unwillingness to accommodate precautions

**Table 2 jcm-12-01648-t002:** Resource Needs Identified by CLL Caregivers During the COVID-19 Pandemic (as of March 2022).

Resource Need	Description	Caregiver Responses
COVID-19 Risk Information (*n* = 83)	Expert information addressing COVID-19 risk with targeted information specific to CLL patients: how the virus will affect them, what to do if they test positive, treatment options, antigen tests to take, and resources to share with family to explain their risk.	Specific info on how COVID impacts the CLL patient—are they even more susceptible now to even the simplest bacterial infection from a cut or scratch, or from any virus they might encounter.
		Something in print to show family and friends WHY we are not meeting up with them and having them over. They don’t understand.
Vaccination Information/Access (*n* = 60)	Information about the vaccine: efficacy, medicines that might affect efficacy, how much immunity someone with CLL has, eligibility for CLL patients, treatment options when vaccines is ineffective, and when to receive the vaccine after immunotherapy.	Vaccine effectiveness for the different types of CLL and medications.
Safety/Precaution Information Including Equipment Access (*n* = 50)	Information about and access to cautionary measures (safety recommendations for patients with CLL): guidance regarding disinfecting surfaces, how to safely interact with immunocompromised people, access to masks and sanitizers, and websites indicating where to get equipment.	More what to do not to do to be safe. Kind of had to figure out initially on our own that vaccinations were not going to be effective for my wife based on medication Ibrutinib she was taking.
Objective, Reliable, Frequent/Ongoing Information (*n* = 35)	Desire for constant flow of information from objective, reliable, truthful, and less politicized and scientific sources to stay current.	I recently just discovered that CLL patients were eligible for a second booster. I’m frustrated because I’m very informed otherwise. This information fell through the cracks.
Information/Access to Monoclonal Infusions (*n* = 24)	Information on monoclonal antibody treatment: types of infusion (e.g., EVUSHELD and Remdesivir), where to find it, and how to access monoclonal infusions.	Wish he could have had monoclonal antibodies but these were not available at this time.

## Data Availability

The datasets generated during and/or analyzed during the current study are available from the corresponding author upon reasonable request.

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
