# Peer review of "Understanding the Impact of COVID-19 on Chronic Lymphocytic Leukemia (CLL) Caregiving and Related Resource Needs"

_jcm, 2023, doi:10.3390/jcm12041648_

Round 1

Reviewer 1 Report

The article titled "Understanding the impact of COVID-19 on Chronic Lymphocytic Leukemia (CLL) caregiving and related resource needs" by D. Bagautdinova et al. neatly investigates the impact of COVID-19 on CLL caregivers based on open-ended surveys and interviews. The authors have used inductive and deductive coding approaches to detect new themes and compare with existing ones and categorize the results based on 2 Aims. The results for Aim 1 (challenges faced by CLL caregivers) and Aim 2 (COVID-19 resources caregivers needed) are presented nicely under the observed themes. While the authors have done a great job in putting everything together in a coherent manner, some pointers can be considered/implemented as follows. This article can be accepted for publication upon addressing the following comments.

1. While the results are neatly presented as different themes under 2 different aims, I find the description on analysis to be lacking and monotonous. I feel the authors could describe the analysis steps and method in a more detailed manner - is it possible to provide a flowchart depiction or any other form for this?

2. This comment is in continuation with the previous one - the thematic analytical steps of CCM in analysis is unclear. (In lines 129-138). Would like for the authors to explain on what Owen's criteria is? Rewrite the statement within braces in ln 136 in a better way. How was the saturation fixed at 40%- please provide reasoning.

3. In Sections 3.1 and 3.2, could you please explain what the average value is and what the range is. For ex, in lns 163 and 171, the values inside and outside of braces are not clearly explained.  

Author Response

Point 1: While the results are neatly presented as different themes under 2 different aims, I find the description on analysis to be lacking and monotonous. I feel the authors could describe the analysis steps and method in a more detailed manner - is it possible to provide a flowchart depiction or any other form for this?

Response 1: Thank you for your feedback. We have updated the Methods section to describe the thematic analysis process from the CCM approach in more detail. We also included a visual depiction of that process (Figure 1).

Point 2: This comment is in continuation with the previous one - the thematic analytical steps of CCM in analysis is unclear. (In lines 129-138). Would like for the authors to explain on what Owen's criteria is? Rewrite the statement within braces in ln 136 in a better way. How was the saturation fixed at 40%- please provide reasoning.

Response 2: Thank you for this suggestion. We further explained saturation and Owen’s criteria as well as that the 40% is what was established (the extent) and not fixed. We’ve also added more information on this and a relevant reference: Hennink, M., & Kaiser, B. N. (2022). Sample sizes for saturation in qualitative research: A systematic review of empirical tests. Social science & medicine (1982), 292, 114523. https://doi.org/10.1016/j.socscimed.2021.114523

Point 3: In Sections 3.1 and 3.2, could you please explain what the average value is and what the range is. For ex, in lns 163 and 171, the values inside and outside of braces are not clearly explained.  

Response 3: Thank you for identifying this. We have revised this section to enhance clarity.

Reviewer 2 Report

The manuscript by Diliara Bagautdinova et al explores the impact of COVID-19 on CLL caregivers across 45 U.S. states, based on a survey to which they got 575 responses and 12 interviewed. The authors identified some recurrent challenges in caregiver responses. Interestingly, coping with distress and living in isolation were the two first challenges, far ahead of losing in person care opportunities and increasing caregiving burden. As a majority of participants are white and most of those participating to interview have no children, these response suggest a predominant effect of familial structure related distress. Nevertheless, losing care opportunities was another key response together with the need for more information.

This is an interesting study with several well-identified bias. A missing information is the severity of the disease as CLL goes from asymptomatic chronic lymphocytosis without associated hypogammaglobulinemia – actually the most common disease form – that does not deserve any specific treatment to severe forms of the disease that deserve treatments. The fraction of patients who deserved a treatment, the disease impact on treatment administration and follow-up, the rate of these patients needing CLL treatment as well as the rate of those requiring EVUSHELD are not described not even discussed. Therefore, I would suggest to better stratify the analysis according to disease severity and treatment.  

Author Response

Point 1: This is an interesting study with several well-identified bias. A missing information is the severity of the disease as CLL goes from asymptomatic chronic lymphocytosis without associated hypogammaglobulinemia – actually the most common disease form – that does not deserve any specific treatment to severe forms of the disease that deserve treatments. The fraction of patients who deserved a treatment, the disease impact on treatment administration and follow-up, the rate of these patients needing CLL treatment as well as the rate of those requiring EVUSHELD are not described not even discussed. Therefore, I would suggest to better stratify the analysis according to disease severity and treatment. 

Response 1: Thank you for your observations on disease severity. We agree it is an important issue. The survey respondents did provide data on their loved one’s current treatment status, including whether they had previous treatment experience and were in remission or had stopped treatment (but not in remission). They also indicated if they were not in treatment (and had not yet had any treatment) and instead were being monitored in “watch and wait”. We’ve added this information to the demographic information on survey respondents to more fully characterize caregivers’ experiences. We’ve also re-analyzed the survey responses and confirmed that all four themes reported in the survey results did emerge regardless of treatment status, which does illustrate the impact of living with CLL during the pandemic, regardless of having severe forms of the disease requiring treatment. Two challenges were notably less prevalent from caregivers of a loved one in watch and wait (care burden and less in-person care opportunities), which we’ve also highlighted in the results. We added in those n sizes to each theme as well to better illustrate the prevalence of those challenges based on whether they were caregivers of patients with current or previous treatment experience or those presumably asymptomatic (i.e., no treatment experience and in watch and wait). The interviews were conducted in April-May 2022, during the time period in which EVUSHELD was first being disseminated to blood cancer patients. We have no clinical data on whether their spouse’s clinician required it, only caregivers’ shared experiences of seeking and obtaining it.

Round 2

Reviewer 2 Report

No additional comment or suggestion